

# EuBrewNet – A European Brewer network (COST Action ES1207), an overview.

John S Rimmer[1], Alberto Redondas[2], Tomi Karppinen[3]

[1]School of Earth and Environmental Sciences, University of Manchester, Manchester, M13 9PL, UK
[2]AEMET, Izaña, Tenerife, Spain
[3]Finnish Meteorological Institute, Arctic Research Center, Finland
*Correspondence to*: John S Rimmer (john.s.rimmer@manchester.ac.uk)

**Abstract.** COST Action ES1207, EuBrewNet, was proposed to coordinate Brewer Spectrophotometer measurements of ozone, spectral UV and aerosol optical depth (AOD-UV) in the UV within Europe, and unite the ozone, UV and AOD communities, through a formally managed European Brewer Network capable of delivering a consistent, spatially homogeneous European data resource, significant for the World Meteorological Organisation (WMO), the World Ozone and UV Data Centre (WOUDC), the International Ozone Commission (IO3C), the Intergovernmental Panel on Climate Change (IPCC), Global Monitoring for Environment and Security (GMES) and the ozone trend assessment panels. Around 50 Brewer Spectrophotometers are deployed in Europe, independently funded by national agencies, each were duplicating effort to achieve separately best practice and accuracy. EuBrewNet was established to remove this disparity, establish knowledge exchange and training, and open up a route to links with international agencies and other networks globally. New instrument characterisation, calibration and consistent data processing algorithms have been developed and applied with a new near real time database providing a range of data products resulting from centralised processing algorithms and quality control methods. Governance of the new network, which has already expanded beyond the boundaries of Europe, will be overseen directly by the WMO Scientific Advisory Groups.



# 1 Introduction

Although our knowledge and understanding of the processes and reactions that affect ozone concentrations in the stratosphere have grown significantly in recent years, there are still uncertainties in the predictions of future trends and the quantification of the effects of policy measures to protect the Ozone Layer. There is also a growing recognition that the

issues of stratospheric ozone depletion and of climate change are closely linked (Hossaini et al, 2015; McKenzie et al, 2011) and that climate change may affect the recovery of the Ozone Layer. Therefore, long-term monitoring continues to be essential to provide the necessary feedback into predictions on the recovery of the stratosphere.

The unprecedented depletion of the arctic ozone layer in spring 2011 (Manney et al, 2011) served as a stark reminder that, 24 years after the Montreal Protocol, our understanding of global and local trends in stratospheric ozone is still far from clear

cut. The corresponding significant increases in UV radiation over large areas of northern Europe were clearly cause for concern. In addition to ozone, aerosols and clouds affect the UV radiation and usefully accurate forecasts also must take account of these factors. The challenge remains to improve the accuracy and understanding of the relationships between UV radiation, ozone, clouds and aerosols. Furthermore, although the ozone variations have been considered mainly for their effect on the ultraviolet radiation, it should be kept in mind that ozone, even if in a small abundance in the atmosphere, plays

a key role in the energy balance of the planet through its involvement in radiative processes. Ozone changes may have lasting consequences within the climate system.

While satellites are routinely used to retrieve atmospheric data products, their accuracy is underpinned by ground station measurements. Once launched, drifts in calibration or errors due to snow or cloud albedo, can only be detected by comparison with ground station data. Spectral UV irradiance products derived from satellite instruments are entirely

estimated, based on radiative transfer models and the retrieved total ozone column (TOC), and they are far from representing the actual radiation field at a specific ground location, particularly under cloudy conditions or at heavily polluted environments. As is particularly true whenever long-term trends are of interest, the fundamental responsibility to define the limits of accuracy therefore rests with the ground station instruments.

The fully automated Brewer Spectrophotometer (Kerr et al, 1985) has provided high quality TOC data for more than 30

years and is now deployed at most of the ground based TOC monitoring stations. It is also capable of measurements of ozone vertical profiles (Umkehr method), spectral UV radiation and aerosol optical depth in the UV (AOD-UV), as well as columns of other trace constituents such as sulphur dioxide and nitrogen dioxide. There are over two hundred Brewers deployed throughout the world, independently operated by national agencies, of which around fifty are located within Europe (some already since the early 1980s). This represents not only a significant proportion of the total global monitoring

effort, but also an extremely valuable resource of co-located TOC, UV and AOD-UV measurements which was being considerably underused due to the lack of coordination and harmonisation between the respective agencies. The co-location





of these measurements is crucial for providing consistent data for research into radiative transfer and forecasting models, however any disparity serves to severely restrict the overall utility. Since Brewer measurements constitute a long term monitoring operation and the data provides the foundation for end users including forecasting agencies, policy decision makers, general public, academic personnel and other researchers, COST Action ES1207 was the ideal, if not the only

mechanism to facilitate the harmonization of procedures and therefore spatially consistent data, through networking and capacity building.

Currently, each monitoring agency is funded nationally to comply with the terms of the Vienna Convention. However, the disparity arises since each station may pursue differing measurement schedules which may not contribute to a coordinated outcome. Furthermore, data processing methods vary such that different agencies may arrive at different results from the

same raw data file. Effects of instrument characteristics on the derived products may be handled differently if, indeed, they are handled at all and quality control methods also vary from site to site. In addition, there have been no protocols to govern updates to software or experimentally determined physical constants. Although the Regional Brewer Calibration Centre – Europe (RBCC-E) has been in place providing support for a first generation calibrated Brewer reference triad on TOC for several years, there are no mandatory and clear protocols for frequency and retro-application of calibration data. The well-

reputed services of the World Ultra-Violet Calibration Centre (WUVCC) have also been underused with too few stations participating to guarantee the homogeneity of time series on spectral UV. In short, there has been no formal European Brewer Network and related regional data base capable of providing spatially consistent data to a high degree of accuracy with a common scale of quality assurance. We therefore need to present the European Brewer stations globally as a formal network with clear operational and data protocols, encompassing the RBCC-E reference triad (one of only two in the world,

the other supported by Environment Canada), which will be a keystone in global monitoring activities and a major step towards a true global network for ozone UV and AOD-UV..

## 2 Characterisation and calibrations

A first step in ensuring consistency is to set standards for observational quality. Individual instruments must therefore be characterised to establish how component characteristics, e.g. slit function, wavelength calibration  (Redondas et al, 2014,

Redondas et al 2017) ,  Photomultiplier  tube (PMT) linearity (Fountoulakis et al, 2016), spectral responsivity, temperature dependence (Berjón et al, 2017; Fountoulakis 2017), stray light (Karppinen et al, 2015, Puli et al 2016, Redondas et al 2017), field of view, angular response, polarisation (Carreño et al, 2016) etc., uniquely affect measurement outcomes and how potential errors may be avoided or corrected for.

In addition, measurement accuracy is ultimately dependent on regular and accurate calibration. Methodologies which ensure

the best transfer of calibration constants and traceability to the reference laboratories, these being the RBCC-E triad for total



ozone, the World Ultra-Violet Calibration Centre (WUVCC) and the Quality Assurance of solar Spectral Ultraviolet irradiance Measurements carried out in Europe (QASUME; Gröbner and Sperfeld, 2005) for the UV measurements, either have been or are being developed. Regular comparison with the Canadian Brewer Triad is also maintained to ensure global compatibility of TOC measurements (León et al 2017).  In order to ensure a traceable laboratory characterization of the

reference Brewer instruments, the reference RBCC-E (Regional Brewer Calibration Center for Europe) Brewer from the calibration triad from the Spanish Meteorological Agency at Izaña and the travelling reference Brewer #158 from the manufacturer Kipp & Zonen were thoroughly characterized for temperature dependency, wavelength bandwidth and stray light in 2016 and 2017 respectively on the frame of the ENV59 Traceability for atmospheric total column ozone (ATMOZ) project. The instruments were characterized using a dedicated climate chamber and tunable laser facilities at the National

Metrology Institutes. The results confirmed the existing understanding and correction of the temperature dependence of Brewer instruments to properly estimate and to reduce the uncertainty of ozone retrieval with Brewers (Redondas 2017, Berjon et al 2017). Also a preliminary error estimation was developed with (Egli 2016) with the objective to implement on the Eubrewnet database.


## 3 Central data processing

For each ozone value five quasi-simultaneous observations are made. Each observation consists of simultaneous measurement of intensity at five UV-wavelengths. Wavelengths are instrument specific but roughly 306.3, 310.1, 313.5, 316.8, and 320.1 nm.  Intensities are written to a raw file for further processing. Raw data is transferred automatically every

half hour from each Brewer on the network to the new EuBrewNet database, hosted by the Agencia Estatal Meteorologia (AEMET).

The algorithm to retrieve TOC from individual observations is based on differential absorption of ozone at the measured wavelengths.  A weighted double ratio, MS9, of the measured intensities is calculated and compared to the similar extra-terrestrial one, ETC, determined by transfer from a travelling reference instrument. The weighting coefficients are designed

to minimize the effects of aerosols and $SO_2$ spectral absorption. A differential absorption coefficient, $\alpha$, calculated from ozone absorption cross sections convoluted with the slit functions at the measurement wavelengths, is used to transform this difference into total ozone on the light path or slant column. To get the vertical total ozone column, $O_{30,}$ the value is further divided by air mass factor $\mu$ (Eq 1).



$$O_{30} = \frac{MS9 - ETC_0}{\mu\alpha} \tag{1}$$

The next step is to apply corrections to the level 0 data, $O_{30}$, based on the instrument characterisation where available. A current controlled halogen lamp located inside the Brewer is used to track instrument response stability. Measurements of this lamp, R6, are compared to a reference, $R6_{ref}$, taken at the time of calibration and used to apply a correction $\Delta_{SL}$ to the

5   measured ozone data (Eq 2).

$$\Delta_{SL} = \frac{R6_{ref} - R6}{\mu\alpha} \tag{2}$$

A series of neutral density filters are present in the input optics of the Brewer and these are characterised for non-linearities, $ETC_N$, so that a filter dependent correction, $\Delta_{Filter}$, can also be applied (Eq 3).

$$\Delta_{Filter} = \frac{ETC_N}{\mu\alpha} \tag{3}$$

Finally, stray light correction is applied to the single Brewers where this has been characterised. The characterisation is based on an exponential fit between single and double Brewer measurements (Karppinen et al, 2015). The correction is an iterative process resulting in bringing the single Brewer values into agreement with the double Brewers, as shown in the example of Figure 1, even at low solar elevation angles (Eq 4).  N in Eq 4 is the iteration index.

15 $$\Delta_{StrayLight,N} = \frac{A * (\mu O_{3,N-1})^B}{\mu\alpha} \tag{4}$$

The constants A and B are determined during the characterisation process. The resulting ozone value is given by combining Eqs 1-4.

$$O_3 = O_{30} + \Delta_{SL}O_3 - \Delta_{Filter}O_3 - \Delta_{StrayLight}O_3 \tag{5}$$

20





Finally the data is filtered to select only valid measurements. For the ozone product, the standard deviation of the 5 quasi-simultaneous ozone measurements must be less than 2.5 Dobson units (DU), the air mass factor must be less than 3.5 (this can be set higher for MKIII double Brewers or where stray light correction has been applied), there must be a valid mercury lamp wavelength calibration before and after the measurement and the measured ozone value must be between 100 DU and

500DU (although this may be edited by the data provider if necessary, e.g. 600 DU may sometimes be observed in Sodankylä). The AOD-UV data product is described elsewhere (López-Solano et al, 2017) and will be implemented within EuBrewNet in the coming months. Similarly work is underway to develop the calibration protocols and processing for the UV data (Lakkala et al, 2016).

## 4 The EuBrewNet near real time database

Figure 2 is a recent snapshot of the EuBrewNet network which is constantly growing. A real time updated version can be found here http://webciai2new.aemet.es/eubrewnet . The core of EuBrewNet is a data storage and scientific information processing system for the Brewer spectrophotometers (http://rbcce.aemet.es/eubrewnet/brewer/index) to which the Brewer raw data is automatically transferred on a half hourly basis. The data is processed, subjected to quality filters, and corrections are applied all at the database in near real time. In this way, all Brewer data from all stations are processed and quality

assured the same way, removing all associated inconsistencies that may otherwise apply across the network. EuBrewNet is closely associated with the Regional Brewer Calibration Centre for Europe (RBCC-E) and this allows the characterisation of instrument specific non-linearities or other sources of error that can be corrected for in the centralised processing. A schematic of the data stream and processing is shown in figure 3.

The calibration and characterisation data for each instrument must also be stored in the database which will then allow the

raw data to be converted into data products in near real time (NRT). The raw data is also uploaded to WOUDC for long term data archiving and back up. The output of ozone products is now operational and these are accessible from the EuBrewNet database. A similar methodology is being developed for the Brewer UV and aerosol optical depth products which should be available in the near future.

The data products are produced as different levels:

**Level 0:** Raw data from the Brewer. This is the unprocessed data which is only available to the providing operator so that appropriate diagnostic checks can be made to check data on a regular basis.

**Level 1.0:** Basic values from calibration data calculated using Eq 1. This is the most basic ozone product without any QA or corrections applied, equivalent to the calculation made by the Brewer control software on site. This enables a comparison to be made to check that the correct calibration data is being used



**Level 1.5:** NRT data changeable over the first week. Calibration and characteristic corrections applied as in Eqs 2-5. This is the first data product available to the registered users. The data is passed through a series of filters, and corrections are applied based on the instrument characterisation and the stability checks by the internal standard lamp. The standard lamp correction is applied using a triangular weighted smoothing 3 days before and after the day of the measurement. This means that L1.5 data can change as more QA information becomes available. This NRT data is most useful for assimilation into forecast models.

**Level 1.6:** Interim data with calibration and characteristic corrections applied. Available to users, this is simply L1.5 data once the standard lamp correction has been fully applied and the value is now stable. This data is most useful for the interim values normally used for day to day reporting.

**Level 2.0:** This is the final processed data for archiving, interpolated over a calibration cycle and also available to users. To comply with WMO best practice, Brewers should be calibrated every two years. The procedure is to first check the status of the calibration, second do any maintenance and lastly set the final calibration. If the initial status of the calibration does not agree with the final calibration of the previous intercomparison, this indicates instrument drift and the ozone values must be re-calculated based on the interpolation between the two points. The resulting L2 data is the highest quality which can then be archived and used for trend analysis.

In addition to levels, the database also stores multiple versions, each containing its own levels as described above. For example, Version 1 contains ozone data using Bass and Paur ozone cross sections. However, the WMO SAG Ozone has recently ruled that the new Bremen cross sections should be used. EuBrewNet makes this easy by re-processing all the data in the data base using the new cross sections and storing it as Version 2, also retaining Version 1, in parallel, for the historical record.

The idea is that all these data products will be available directly via a link with the WOUDC so that users do not need to go to a different data base. However, this link is still under construction under the supervision of a sub-group of the WMO SAG-Ozone. For the moment, users may register to access data at http://rbcce.aemet.es/eubrewnet/brewer/index , which includes a wiki which contains information about the system and user support, and further information including instructions on how to contribute to the network can be found at www.eubrewnet.org .

## 5 Regional Calibration Centre for Europe (RBCC-E)

The RBCC-E campaigns and regular calibration play an important role on the final data (Level 2), as is indicated in equation 2 , the operative ozone is corrected by the Standard Lamp,  the standard lamp reference value is provided during the





calibration and examined during the intercomparison campaign (Redondas 2017). The previous calibration and new calibration are compared with and without the Standard Lamp correction to assure that the internal standard lamp is tracking the changes on calibration.

As a result of the calibration the validity of the current calibration is examined during the campaign and if a new calibration
is provided, the history of the instrument is studied to determine from when the observations have to be reprocessed using a step function or if the changes are continuous and linear evolution of the instrument calibration are implemented between calibrations. In both cases comparison with external instruments, neighbouring Brewer or satellite overpass measurements, will help to evaluate the application of these functions.

The campaigns also help on the characterization of the instruments, the comparison with a well maintained and characterized
instrument reveals instrumental characteristics that are difficult to detect at the station without a reference. In particular the filter correction (Figure 2) and the Stray light (Figure 1)

The introduction of the instrumental characterization greatly improves the results of the intercomparison (Redondas and Rodriguez, 2012), in particular the Stray light correction application at the final calibration brings all the instrument to the - +/- 0.5 % range (Redondas 2017) on the full range.

**6 Conclusions**

COST Action ES1207, "EuBrewNet – A European Brewer Network", was awarded to allow the harmonisation and coordination of Brewer ozone spectrophotometer measurements of TOC, spectral UV and AOD-UV. New instrument characterisation methodologies have been implemented and calibration campaigns have been carried out which use these new methodologies to greatly improve the accuracy of the results. Common data processing and quality assurance also now
ensure consistency of measurements throughout the network.

A major part of EuBrewNet is the new NRT database which automatically collects raw data from the instruments and applies the new data processing and quality assurance centrally, thereby ensuring the consistency of the resulting data products across the network. Currently, the TOC processing is in operation and NRT data is available to registered users. The AOD-UV processing algorithms have been developed but are yet to be implemented. Similarly the UV processing is still under
development but should be implemented in the early part of 2018.

A link to the WOUDC is currently under construction so that data will be available through a portal at the WOUDC web site. The Brewer raw data is also transferred to the WOUDC for long term archiving. Governance of the EuBrewNet is to be overseen by the WMO SAG Ozone.

Further details of EuBrewNet and registration for data are available at the web site www.eubrewnet.org.



**Acknowledgements**

The authors would like to acknowledge networking support by the COST Action ES1207, and contributions from the Regional Brewer Calibration Centre for Europe and the World Radiation Centre. The authors also acknowledge the considerable support from all the working group members of COST Action ES1207 who contributed significantly to the
project.

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





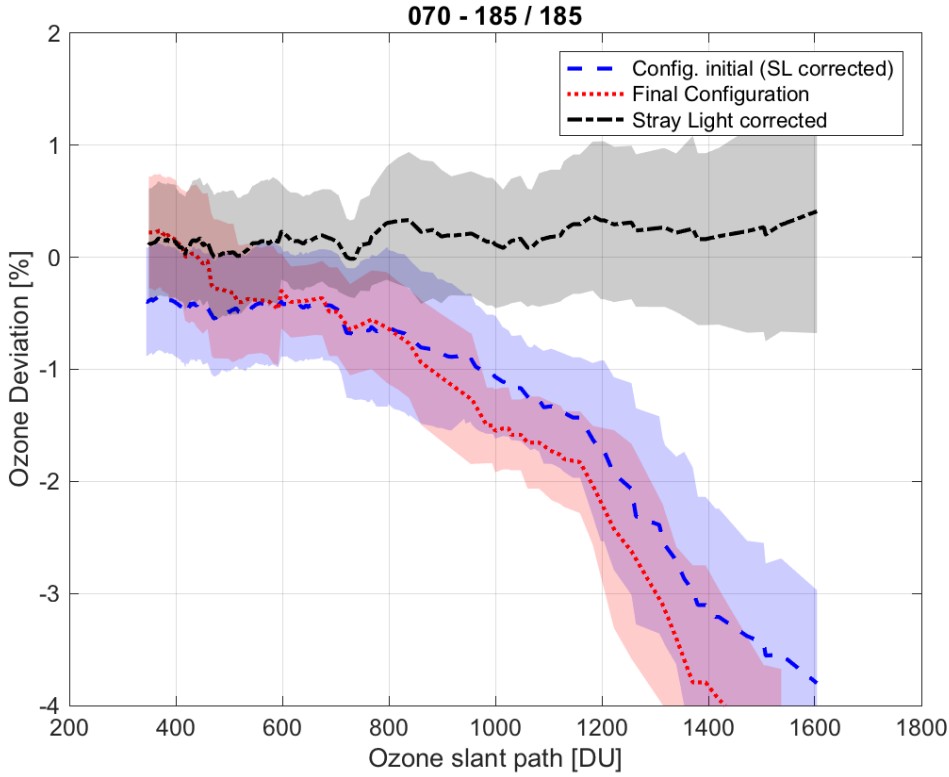

**Figure 1: A comparison of a single Brewer, #070, with the RBCC-E reference Brewer, #185, showing stray light correction. The blue and red lines are the initial ozone deviation of #070 from the reference before and after calibration. The black line shows the stray light corrected values.**




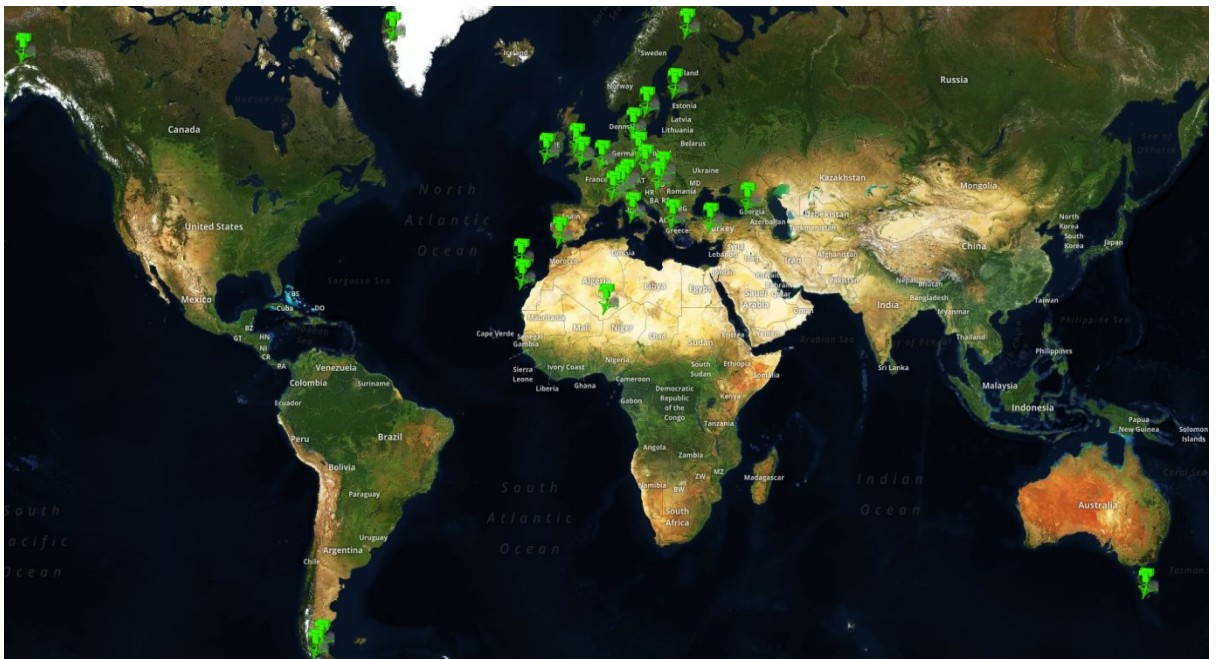

**Figure 2: A snapshot of the Brewer stations contributing to EuBrewNet.**





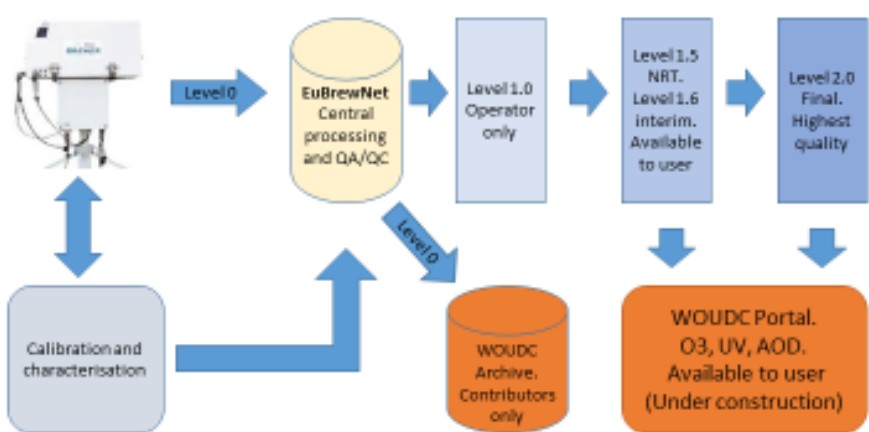

**Figure 3: Schematic of the EuBrewNet near real time database.**