# Peer review of "EuBrewNet – A European Brewer network (COST Action ES1207), an overview."

_Atmospheric Chemistry and Physics, 2017_

## Referee Comment (RC1) · Anonymous Referee #1 · 13 Feb 2018

EuBrewNet - A European Brewer network (COST action ES1207), an overview by J. Rimmer et al.

The manuscript describes the progress made during the Cost action in establishing a quality assessed and quallity controlled network of Brewer instruments in Europe, with the focus on total ozone column (TOC) measurements. This is important and relevant for a broad audience interested in high quality ozone measurements (eg. for trend analysis or validation of satellite observations). Therefore it would be worth publishng in ACP. There are however some points that need improvement before acceptance which I list hereafter.

The introduction should make more references to relevant papers. Some examples:

[Figure]

p2 near line 15: give a reference for the involvement of ozone in radiative processes

p2 near line 22: give reference about representativity of satellite measurements

p2 near line 24: refer also to the Brewer manual

The description of the procedures for calculation of TOC and corrections could be better clarified. This would help the non Brewer specialist.

p4 line 17: The description should be corrected: actually the Brewer instrument makes 5 consecutive measurement (direct sun observations). Each of these consists of a number of cycles (standard 20) of quasi simultatious measurements of the different wavelengths by fast switching the entrence slit mask.

p4 line 23: it should be clear which double ratio is used. Maybe it is better to write down explicitly in a general notation (eg see De Backer and De Muer, JGR 96, D11, p20711-20719, 20 Nov 1991) what is meant actually. MS9 is a naming of an internal variable of the Brewer software and it is the log of the measured intensity ratio.

p5 line 6: if the notation above is adopted it becomes clear that R6 is the same measurement but on the internal standard lamp

on p5 line 19: is this formula correct? what is the meaning of the multiplication factor O3 on the right hand side of eq (5)? please clarify.

on p6 line 1-2: it is more accurate to mention this as 5 consecutive measurements (see also comment above p4 line 17)

on p 8 line 11 there is reference to a figure 2 that mentions filter correction, but there is no figure about the filter correction; please correct

---

## Referee Comment (RC2) · Anonymous Referee #2 · 16 Feb 2018

General Comments

The manuscript provides an overview of the achievements made (and those still promised) of the "EuBrewNet" network as funded by a specific COST action. The project has been very important for improving the quality, consistency and transparency of Brewer data in Europe and beyond.

This is of great interest to users of high quality total ozone data and lies within the scope of ACP.

My general comment is that in some places the manuscript currently reads more like a project proposal than a scientific paper. There are a number of places where I feel the details could be tightened up or the marketing talk toned down.

[Figure]

Once this has been done I believe it will be suitable for publication in ACP.

Specific Comments

Lines 8-20 I think the whole abstract should be rewritten to be more specific about the contents of the paper and less polemical.

Lines 11-13 This list of relevant international bodies sounds like a project proposal or funding pitch and I think should be deleted.

Lines 2-17 These statements should be better referenced. I suggest the UNEP/WMO Ozone Assessments as authoritative references for many of these claims. Hossiaini et al. 2015 seems a random choice to choose for this topic.

Line 8-10 You can't say the 2011 Arctic depletion showed a lack of understanding, because CTMs were able to reproduce the event forced with the actual meteorology, but it did illustrate that low ozone and high UV can still be a big issue in Europe despite the success of the Montreal Protocol. You could refer to recent work on ozone trends (eg Chipperfield et al. 2017, Weber et al. 2018).

Lines 17-23 Again I think these statements should be referenced.

Line 24 "TOC" has not previously been defined.

Line 25 For the information of readers not as familiar with the subject, you should also mention the Dobson which is slowly being supplanted by Brewers. I am not sure "most" is correct globally but no doubt it is true in Europe these days.

Line 30 "was being" - before what?

Line 2 "any disparity" – disparity in what? (I assume you mean disparity in technique).

Line 2 – 6 This statement is too hyperbolic for a scientific paper – again it sounds like a proposal.

Line 7 I would drop or re-work this sentence. The separate funding is not the issue from a scientific point of view, the real issue is the different schedules, processing etc which you then go on to describe.

Lines 8-10 I would prefer more detail here. How does a different schedule affect the measurements? Were the results of different processing significantly different from each other? How big an effect do the instrument characteristics make?

Line 13 – Explain what you mean by "first generation".

Lines 18-22 Now the writing has changed to future tense. This again makes the manuscript sound like a proposal.

Lines 23-28 This paragraph is excellent because it lists the specific issues and gives references for each. However I don't like the "etc" because either EuBrewNet characterises these properties or does not. You could give a reference for slit functions too.

Line 9-10 "the National Metrology Institutes" – which ones?

Line 12 "developed with" should be be just "developed" unless there was going to be something else in the sentence.

Line 18 Personally I think "roughly" is too informal for a journal paper and would prefer "approximately".

Line 23 I think it would be better to express these quantities (such as MS9 and O30) in more general notation (in fact as it normally appears in Brewer papers) and then give their equivalent in Brewer-specific terminology.

Line 1 ETC_0 hasn't been defined yet.

Line 2 This is perhaps a philosophical discussion but I am surprised the first derivation of ozone is considered level 0 data. Level 0 would normally be the raw intensities. To calculate ozone you need to have an algorithm for mu and alpha and these have previously changed, and will continue to change, for example with new cross-sections.

Line 7 How are the filters characterised for non-linearities?

Line 11 "stray light correction" should be "a stray light correction"

Line 15 It should be explained more clearly when these iterations are performed. By the notation it appears O3_0 -> O3_1 -> O3_2 etc going up in the processing levels.

Line 19 O3 is not meant to be on both sides of the equation, is it?

Line 1 "Finally the data is filtered to select only valid measurements" – this makes it sound as if your filters are 100% accurate in removing all bad data but no good data. Maybe re-word to something like "to try to select only valid measurements".

Lines 10-18 The arrangement of the paper seems wrong here because we have already read about the central processing in detail in the previous section but now we are being introduced to it again in general terms. (I suspect there has been cutting & pasting from different co-authors' contributions). Please ensure the different sections are unified and flow together properly.

Line 17 It's not really correct to say the "WMO SAG Ozone has recently ruled...", in fact the International Ozone Commission wrote to the SAG directing them to implement the new cross sections (and to take stratospheric temperature into account too).

Line 11 Figure 2 is nothing to do with the filter correction. The filter correction was previously alluded to but has remained somewhat mysterious to the reader.

Line 12-14 This is important. It is good to see the effect of all your work quantified like this. I would have liked to see more quantitative detail like this throughout the rest of the manuscript.

Technical comments

Page 2 Line 4 and Line 6 - It is unusual for "Ozone Layer" to be in capitals.

Throughout, "et al" should be "et al. "

The spellings of words are inconsistent, both "characterise" and "charcaterize" are used in different places, presumably by different co-authors.

References

Chipperfield, M. P., Bekki, S., Dhomse, S., Harris, N. R. P., Hassler, B., Hossaini, R., Steinbrecht, W., Thiéblemont, R. & Weber, M. (2017). Detecting recovery of the stratospheric ozone layer. Nature, 549(7671), 211–218. https://doi.org/10.1038/nature23681

Weber, M., Coldewey-Egbers, M., Fioletov, V. E., Frith, S. M., Wild, J. D., Burrows, J. P., Long, C. S., and Loyola, D.: Total ozone trends from 1979 to 2016 derived from five merged observational datasets – the emergence into ozone recovery, Atmos. Chem. Phys., 18, 2097-2117, https://doi.org/10.5194/acp-18-2097-2018, 2018.
* * *

---

## Referee Comment (RC3) · Anonymous Referee #3 · 26 Feb 2018

This paper provides a concise overview of the motivation for and implementation of the EuBrewNet activity. However, it falls short in summarizing the breadth of specific early achievements and in discussing EuBrewNet progress in developing improved linkages to other agencies and networks. This project is, indeed, a major step towards achieving a quality-assured uniform international database for ozone, spectral UV, and aerosol optical depth from Brewer measurements. However, consideration should be given to revising the manuscript so as to acknowledge already existing efforts that this action builds upon and to provide an indication of the road forward beyond EuBrewNet. As written, a reader without extensive knowledge of existing measurement activities (either isolated or coordinated within established networks) could get the impression that such measurements have been in such disarray as to be useless for scientific

trends and process studies. Further, while I am a strong supporter of EuBrewNet. I do not think that it will solve every problem (as seems to be indicated) but rather will point to the next steps that must be taken. Specific page-by-page comments follow.

Page 1, lines 15-20: The inclusion of more details on the specific achievements to date in these areas would make this a much-improved paper.

Page 2, lines 3-4: It is incorrect to state that there are uncertainties regarding the effects of ozone protection policy measures. The efficiency of the Montreal Protocol with respect to protecting the ozone layer from depletion by halocarbons is well understood and documented. The combined effects of the declining influence of chemical depletion and the increasing influence of climate change complicate the prediction of future ozone trends. Indeed, this is mentioned. However, the way it's presented makes it sound like we don't have a handle on the chlorofluorocarbon issue.

Page 2, lin6 6: Suggest changing the wording to "will influence the evolution of the ozone layer".

Page 2, lines 8-9: This statement is simply not true! The possibility of severe Arctic ozone depletion, such as occurred in spring 2011, was stated following the results obtained from airborne campaigns conducted during 1989-1992. Substantial ozone loss was projected to occur in years when low vortex temperatures persisted into late February and beyond. Our understanding of the chemical depletion processes is quite robust.

Page 3, lines 4-5: The COST action is a great mechanism for facilitating harmonization and quality assurance in Brewer measurement. However, to state that it is the only mechanism is somewhat of an overstatement. There are efforts in existing networks to achieve similar results. For example, the Dobson/Brewer Working Group of the Network for the Detection of Atmospheric Composition Change has developed specific protocols for such work and the investigators are involved in EuBrewNet.
Page 3, lines 7-21: Admittedly there has been a lack of uniformity and standardization in Brewer measurements. However, are there no examples of stations at which experienced investigators have been conducting measurements and analyses "properly"? If so, would it not be appropriate to cite some examples and then discuss how Eu-BrewNet will amplify such procedures throughout Europe. As presented, the reader is given the impression that previous data from Brewer sites should be viewed with great skepticism.

Section 2: There is no mention in this section of the possible effects of using different ozone cross-sections. In addition, while the ATMOZ project is mentioned, none of the initial results are summarized. Admittedly, there is a reference to (Redondas, 2017). However, the references include two such papers, both of which were submitted very recently. My understanding is that there were some wavelength calibrations issues discovered. Some mention of the results and the path forward would improve this manuscript. In addition, I would have expected a section on characterization and calibration to address how possible comparisons with data obtained using other co-located instrument types might be used for establishing measurement accuracy. Finally, there is no mention of how long-term instrument stability will be verified.

Section 3: While details are provided on the retrieval of TOC, the section does not specifically address how central data processing will actually be implemented throughout the network. The need for valid mercury lamp wavelength calibration is stated; however, specific details or recommendations for such calibrations are not provided.

Section 4: The implementation of a near real time database will be an important aspect of EuBrewNet. However, unless provisions are made for some preliminary scientific analyses of the results by someone (i.e., to ascertain whether the data make sense from a geophysical point of view) there is a risk that erroneous data could be posted. Having two versions of the level 2.0 data corresponding to the use of two different sets of cross sections can be quite valuable when trying to intercompare with data obtained outside of the network or when attempting to generate a merged data set. Are there no

**ACPD**
results that can be shown on the effect of using one or the other set of cross sections?

Section 5: Is there a path forward suggested by the results from the recent intercomparison campaign. There is a reference given; but the paper has just been submitted.

Section 6: This manuscript could be improved considerably if it included more specific details to support the achievements listed in this section.

---

## Author Comment (AC1) · 27 Apr 2018

The authors would like to thank all referees for their valuable time and welcome their comments. It is hoped that the following responses will be satisfactory.

Referee 1.

EuBrewNet - A European Brewer network (COST action ES1207), an overview by J. Rimmer et al. The manuscript describes the progress made during the Cost action in establishing a quality assessed and quallity controlled network of Brewer instruments in Europe, with the focus on total ozone column (TOC) measurements. This is important and relevant for a broad audience interested in high quality ozone measurements (eg. for trend analysis or validation of satellite observations). Therefore it would be

worth publishng in ACP. There are however some points that need improvement before acceptance which I list hereafter.

The introduction should make more references to relevant papers. Some examples:

p2 near line 15: give a reference for the involvement of ozone in radiative processes

... *A reference has been added as suggested*

p2 near line 22: give reference about representativity of satellite measurements

... *A reference has been added as suggested*

p2 near line 24: refer also to the Brewer manual

... *The Brewer manual has been referred to and listed in the reference section.*

The description of the procedures for calculation of TOC and corrections could be better clarified. This would help the non Brewer specialist.

p4 line 17: The description should be corrected: actually the Brewer instrument makes 5 consecutive measurement(direct sun observations). Each of these consists of a number of cycles (standard 20) of quasi simultanious measurements of the different wavelengths by fast switching the entrence slit mask.

... *The authors take note of the correction suggested, however the measurements are taken by fast switching of the exit slit mask. The manuscript has been revised as follows: "Each observation consists of 20 cycles of quasi-simultaneous measurements of intensity at five UV-wavelengths by fast switching of the spectrometer exit slit mask."*

p4 line 23: it should be clear which double ratio is used. Maybe it is better to write down explicitly in a general notation (eg see De Backer and De Muer, JGR 96, D11, p20711-20719, 20 Nov 1991) what is meant actually. MS9 is a naming of an internal variable of the Brewer software and it is the log of the measured intensity ratio.

... *The authors appreciate that the description of the algorithm does not go into any*

*great depth. However, this paper is an overview aimed at the Brewer data user rather than the Brewer operator and as such the objective was to keep it simple. We have added references pointing to more in depth descriptions of the standard algorithm and have changed the notation of the double ratio to be more consistent with the internal standard lamp equation. The idea is to make it easier to see how the added data enhancements work.*

p5 line 6: if the notation above is adopted it becomes clear that R6 is the same measurement but on the internal standard lamp

... *See above.*

on p5 line 19: is this formula correct? what is the meaning of the multiplication factor O3 on the right hand side of eq (5)? please clarify.

... *This error is typographic. The O3 is not a factor, it was intended as a label for the delta. The equation has been corrected to: O3 = O30+△O3SL−△O3Filter−△O3StrayLight*

on p6 line 1-2: it is more accurate to mention this as 5 consecutive measurements (see also comment above p4 line 17)

... *Point taken. Quasi-simultaneous has been replaced by consecutive.*

on p 8 line 11 there is reference to a figure 2 that mentions filter correction, but there is no figure about the filter correction; please correct

... *The reference to figure 2 has been removed.*

Referee 2.

General Comments The manuscript provides an overview of the achievements made (and those still promised) of the "EuBrewNet" network as funded by a specific COST action. The project has been very important for improving the quality, consistency and transparency of Brewer data in Europe and beyond. This is of great interest to users

of high quality total ozone data and lies within the scope of ACP. My general comment is that in some places the manuscript currently reads more like a project proposal than a scientific paper. There are a number of places where I feel the details could be tightened up or the marketing talk toned down. Once this has been done I believe it will be suitable for publication in ACP.

Specific Comments

Page 1 Lines 8-20 I think the whole abstract should be rewritten to be more specific about the contents of the paper and less polemical.

... *The abstract has been re-written.*

Lines 11-13 This list of relevant international bodies sounds like a project proposal or funding pitch and I think should be deleted.

... *Have been deleted.*

Page 2 Lines 2-17 These statements should be better referenced. I suggest the UNEP/WMO Ozone Assessments as authoritative references for many of these claims. Hossiaini et al. 2015 seems a random choice to choose for this topic.

... *Reference to the WMO/UNEP ozone assessments has been added.*

Line 8-10 You can't say the 2011 Arctic depletion showed a lack of understanding, because CTMs were able to reproduce the event forced with the actual meteorology, but it did illustrate that low ozone and high UV can still be a big issue in Europe despite the success of the Montreal Protocol. You could refer to recent work on ozone trends (eg Chipperfield et al. 2017, Weber et al. 2018).

... *The sentence has been re-written including the suggested references: "The un-precedented depletion of the arctic ozone layer in spring 2011 (Manney et al, 2011) served as a stark reminder that, 24 years after the Montreal Protocol, our understanding of trends in stratospheric ozone (Chipperfield et al. 2017, Weber et al. 2018) is still*

[Figure]

*important ."*

Lines 17-23 Again I think these statements should be referenced.

... *Included in corrections for referee 1.*

Line 24 "TOC" has not previously been defined.

... *TOC is defined in the sentence "Spectral UV irradiance products derived from satellite instruments are entirely estimated, based on radiative transfer models and the retrieved total ozone column (TOC), and they are far from representing the actual radiation field at a specific ground location (Zempilaetal., 2016), particularly under cloudy conditions or at heavily polluted environments."*

Line 25 For the information of readers not as familiar with the subject, you should also mention the Dobson which is slowly being supplanted by Brewers. I am not sure "most" is correct globally but no doubt it is true in Europe these days.

... *The Dobson has now been mentioned and 'most' limited to Europe.*

Line 30 "was being" - before what?

... *'was being' changed to 'was previously being'*

Page 3 Line 2 "any disparity" – disparity in what?  (I assume you mean disparity in technique).

... *'disparity' changed to 'operational disparity'*

Line 2 – 6 This statement is too hyperbolic for a scientific paper – again it sounds like a proposal.

... *The statement has been revised to "The aim of COST Action ES1207 was to facilitate the harmonization of procedures and therefore provide spatially consistent data."*

Line 7 I would drop or re-work this sentence.  The separate funding is not the issue from a scientific point of view, the real issue is the different schedules, processing etc

which you then go on to describe.

... *Dropped!*

Lines 8-10 I would prefer more detail here. How does a different schedule affect the measurements? Were the results of different processing significantly different from each other? How big an effect do the instrument characteristics make?

... *Examples of the effects of schedules, different processing and characterisations have been given.*

Line 13 – Explain what you mean by "first generation".

... *Explanation included in parenthesis.*

Lines 18-22 Now the writing has changed to future tense. This again makes the manuscript sound like a proposal.

... *This last sentence of the paragraph has been deleted*

Lines 23-28 This paragraph is excellent because it lists the specific issues and gives references for each. However I don't like the "etc" because either EuBrewNet characterises these properties or does not. You could give a reference for slit functions too.

... *'etc' deleted. The slit function is determined by in situ measurement, this has been added in parenthesis.*

Page 4 Line 9-10 "the National Metrology Institutes" – which ones?

... *Explanations have been added*

Line 12 "developed with" should be just "developed" unless there was going to be something else in the sentence.

... *Corrected*

Line 18 Personally I think "roughly" is too informal for a journal paper and would prefer "approximately".

... *Agreed but have substituted 'nominally'*

Line 23 I think it would be better to express these quantities (such as MS9 and O30) in more general notation (in fact as it normally appears in Brewer papers) and then give their equivalent in Brewer-specific terminology.

... *This has been addressed by Referee 1.*

Page 5 Line 1 ETC0 has not been defined yet.

... *Now defined in the second line of the paragraph.*

Line 2 This is perhaps a philosophical discussion but I am surprised the first derivation of ozone is considered level 0 data. Level 0 would normally be the raw intensities. To calculate ozone you need to have an algorithm for mu and alpha and these have previously changed, and will continue to change, for example with new cross-sections.

... *In fact the Brewer does produce and store a raw value for ozone based on some constants hard-coded in the software but in reality this would never be submitted to any data centres for scientific use. In any case, this is the classification in EuBrewNet and this paper can only report it as it is.*

Line 7 How are the filters characterised for non-linearities?

... *The explanation has been added.*

Line 11 "stray light correction" should be "a stray light correction"

... *Corrected*

Line 15 It should be explained more clearly when these iterations are performed. By the notation it appears $O30- > O31- > O32$etc going up in the processing levels.

... *An explanation has been included.*

Line 19 O3 is not meant to be on both sides of the equation, is it?

... *This has been addressed by referee 1.*

Page 6 Line 1 "Finally the data is filtered to select only valid measurements" – this makes it sound as if your filters are 100

... *The sentence has been revised to "Finally the data is filtered to select only those measurements which conform as follows:"*

Lines 10-18 The arrangement of the paper seems wrong here because we have already read about the central processing in detail in the previous section but now we are being introduced to it again in general terms. (I suspect there has been cutting pasting from different co-authors' contributions). Please ensure the different sections are unified and flow together properly.

... *Section 4 has been edited so that details of the processing are not re-introduced.*

Page7 Line17 It's not really correct to say the "WMO SAG Ozone has recently ruled...", in fact the International Ozone Commission wrote to the SAG directing them to implement the new cross sections (and to take stratospheric temperature into account too).

... *This statement has been corrected as suggested*

Page 8 Line 11 Figure 2 is nothing to do with the filter correction. The filter correction was previously alluded to but has remained somewhat mysterious to the reader.

... *Dealt with by referee 1.*

Line 12-14 This is important. It is good to see the effect of all your work quantified like this. I would have liked to see more quantitative detail like this throughout the rest of the manuscript.

Technical comments Page 2 Line 4 and Line 6 - It is unusual for "Ozone Layer" to be in

capitals.

... *Capitals removed.*

Throughout, "et al" should be "et al. "

... *Done.*

The spellings of words are inconsistent, both "characterise" and "charcaterize" are used in different places, presumably by different co-authors.

... *Done.*

References

Chipperfield, M. P., Bekki, S., Dhomse, S., Harris, N. R. P., Hassler, B., Hossaini, R., Steinbrecht, W., Thiéblemont, R. Weber, M. (2017). Detecting recovery of the stratospheric ozone layer. Nature, 549(7671), 211–218. https://doi.org/10.1038/nature23681 Weber, M., Coldewey-Egbers, M., Fioletov, V. E.,

Frith, S. M., Wild, J. D., Burrows, J. P., Long, C. S., and Loyola, D.: Total ozone trends from 1979 to 2016 derived from five merged observational datasets–the emergence into ozone recovery, Atmos. Chem. Phys., 18, 2097-2117, https://doi.org/10.5194/acp18-2097-2018, 2018.

Referee 3

This paper provides a concise overview of the motivation for and implementation of the EuBrewNet activity. However, it falls short in summarizing the breadth of specific early achievements and in discussing EuBrewNet progress in developing improved linkages to other agencies and networks. This project is, indeed, a major step towards achieving a quality-assured uniform international database for ozone, spectral UV, and aerosol optical depth from Brewer measurements. However, consideration should be given to revising the manuscript so as to acknowledge already existing efforts that this action builds upon and to provide an indication of the road forward beyond EuBrewNet.

As written, a reader without extensive knowledge of existing measurement activities (either isolated or coordinated within established networks) could get the impression that such measurements have been in such disarray as to be useless for scientific trends and process studies. Further, while I am a strong supporter of EuBrewNet. I do not think that it will solve every problem (as seems to be indicated) but rather will point to the next steps that must be taken.

Specific page-by-page comments follow.

Page 1, lines 15-20: The inclusion of more details on the specific achievements to date in these areas would make this a much-improved paper.

... *This has been done in the re-write of the abstract suggested by referee 2*

Page 2, lines 3-4: It is incorrect to state that there are uncertainties regarding the effects of ozone protection policy measures. The efficiency of the Montreal Protocol with respect to protecting the ozone layer from depletion by halocarbons is well understood and documented. The combined effects of the declining influence of chemical depletion and the increasing influence of climate change complicate the prediction of future ozone trends. Indeed, this is mentioned. However, the way it's presented makes it sound like we don't have a handle on the chloroflourocarbon issue.

... *This has been addressed by referee 2*

Page 2, lin6 6: Suggest changing the wording to "will influence the evolution of the ozone layer".

... *Done.*

Page 2, lines 8-9: This statement is simply not true! The possibility of severe Arctic ozone depletion, such as occurred in spring 2011, was stated following the results obtained from airborne campaigns conducted during 1989-1992. Substantial ozone loss was projected to occur in years when low vortex temperatures persisted into late February and beyond. Our understanding of the chemical depletion processes is quite

robust.

*... This has been addressed by referee 2.*

Page 3, lines 4-5: The COST action is a great mechanism for facilitating harmonization and quality assurance in Brewer measurement. However, to state that it is the only mechanism is somewhat of an overstatement. There are efforts in existing networks to achieve similar results. For example, the Dobson/Brewer Working Group of the Network for the Detection of Atmospheric Composition Change has developed specific protocols for such work and the investigators are involved in EuBrewNet.

*... This paragraph has already been re-written following comments from previous referees.*

Page 3, lines 7-21: Admittedly there has been a lack of uniformity and standardization in Brewer measurements. However, are there no examples of stations at which experienced investigators have been conducting measurements and analyses "properly"? If so, would it not be appropriate to cite some examples and then discuss how EuBrewNet will amplify such procedures throughout Europe. As presented, the reader is given the impression that previous data from Brewer sites should be viewed with great scepticism.

*... This paragraph has been modified following comments from previous referees, however a further sentence has been added to address this referees concerns over the impression of viewing previous data with scepticism. The authors do not feel it would be right to pick out investigators who are doing it 'properly' as this would be tantamount to denigrating the rest.*

Section 2:

There is no mention in this section of the possible effects of using different ozone cross-sections.

*... Choice of cross sections is not mentioned here as it cannot be decided by Eu-*

*BrewNet. The use of different cross sections is discussed in section 4 in relation to the database and a reference is given on the effects on the data.*

In addition, while the ATMOZ project is mentioned, none of the initial results are summarized. Admittedly, there is a reference to (Redondas, 2017). However, the references include two such papers, both of which were submitted very recently. My understanding is that there were some wavelength calibrations issues discovered. Some mention of the results and the path forward would improve this manuscript.

... *The authors did not intend to discuss the ATMOZ project here. It was mentioned simply to point out that the result was obtained from that experiment and was not part of a EuBrewNet campaign. However, further information has been added.*

In addition, I would have expected a section on characterization and calibration to address how possible comparisons with data obtained using other co-located instrument types might be used for establishing measurement accuracy. Finally, there is no mention of how long-term instrument stability will be verified.

... *These points are addressed in sections 4 and 5. Instrument stability is assessed by the interpolations necessary to produce Level 2 data and the comparison with satellite overpass data and neighbouring instruments can be used to assist with this process.*

Section 3: While details are provided on the retrieval of TOC, the section does not specifically address how central data processing will actually be implemented throughout the network.

... *The authors are puzzled by this comment. Central data processing will be implemented centrally, not throughout the network. A description is given in section 4.*

The need for valid mercury lamp wavelength calibration is stated; however, specific details or recommendations for such calibrations are not provided.

... *The detail has been added.*

Section4:

The implementation of a near real time database will be an important aspect of Eu-BrewNet. However, unless provisions are made for some preliminary scientific analyses of the results by someone (i.e., to ascertain whether the data make sense from a geophysical point of view) there is a risk that erroneous data could be posted.

... *This is a valid point but individual stations have already been making comparisons with overpass data as a check on data validity. It is hoped that further, more in depth investigations can be made as the database matures.*

Having two versions of the level 2.0 data corresponding to the use of two different sets of cross sections can be quite valuable when trying to intercompare with data obtained outside of the network or when attempting to generate a merged data set. Are there no results that can be shown on the effect of using one or the other set of cross sections?

... *Some work has been done here by one of us but outside of the EuBrewNet project. A reference has been added in section 4.*

Section 5:

Is there a path forward suggested by the results from the recent intercomparison campaign. There is a reference given; but the paper has just been submitted.

... *The current work has focussed mainly on TOC. We have added a sentence indicating that the work is not yet complete and we would like to see similar achievements with UV and AOD-UV.*

Section 6:

This manuscript could be improved considerably if it included more specific details to support the achievements listed in this section.

... *The authors would like to say again that this is an overview. The specific details are included in other reports, some also submitted to and/or accepted for publication in this*

*QOS special edition and references have been given rather than deal with copy write over figures etc.. However, a link has also been included to the EuBrewNet WiKi which contains more specific technical information.*

———————————————————